# Implementation of Compassionate Communities: The Taipei Experience

**DOI:** 10.3390/healthcare10010177

**Published:** 2022-01-17

**Authors:** Chia-Jen Liu, Sheng-Jean Huang, Samuel Shih-Chih Wang

**Affiliations:** 1Taipei City Hospital, Taipei City 10341, Taiwan; liu920411@livemail.tw (C.-J.L.); sjhuang1@ntu.edu.tw (S.-J.H.); 2Department of Health and Welfare, Tian-Mu Campus, College of City Management, University of Taipei, Taipei City 111036, Taiwan

**Keywords:** health promoting palliative care, home death, compassionate communities, cultural sensitive, integrated, public-private-partnership

## Abstract

A worldwide movement to empower communities to support their members to care for each other at the end of life (EoL) has emerged since Kellehear published the Compassionate City Charter. This current report discusses the implementation experiences and preliminary outcomes of Compassionate Communities (CC) in Taipei City. Using the guidance of the Charter and international experiences, we have developed and multiplied a culturally sensitive, sustainable, and holistic CC program that composes municipal hospital, social, and other services, partnering with community leaders, non-governmental organizations, university students, and volunteers. Innovative campaigns, such as workshops, conferences, and the Life Issue Café, have been delivered to facilitate engagement, public education, and leadership with reverence to folk beliefs and the use of existing social networks. We have identified a model with strong collaborative leadership, high participation rates, and ongoing commitment. The gaps between asking/accepting and providing help were bridged when social connectedness was strengthened. We also integrated home-based medical care, home-based palliative care, and advance care planning to help the vulnerable who live alone, with poor status, or with limited resource access, and continue to support the community throughout the COVID-19 pandemic.

## 1. Introduction

Many international professionals and scholars have recognized the significance of end-of-life care and grief as a public health issue [1,2,3,4,5]. As a result, a global movement, Compassionate City/Community (CC), has been formed. This paradigm shift is occurring in medical systems. It necessitates community partnerships and community energy to build mutual support, providing a supportive environment for palliative care and grief counseling [6]. Inspired by the Healthy Cities or Healthy Communities program of the World Health Organization (WHO), CC is a concept pioneered by Professor Allan Kellehear of the University of Bradford in the United Kingdom. This public health notion first emerged in the 1980s, and policies and practices in the field of health education can be dated back even further [7,8]. According to this, health is more than just disease-free. The new concept urged the general population to recognize that health is no longer the sole responsibility of doctors and hospitals but everyone. Hospitals can assist patients suffering from accidents, disabilities, or acute or chronic diseases. However, prevention and early intervention are the most effective treatments.

The spirit of this new wave of public health is prevention, harm reduction, and early intervention [7,8]. This health city movement has urged the government, workplaces, schools, and other social organizations to offer relevant policies. Mass media and legislation have discouraged people from using harmful substances or staying in harmful conditions, thereby adopting a healthy lifestyle. Kellehear built the principles of health promotion, particularly the Ottawa Charter (1986), advocating that health promotion and well-being should be complimentary. He integrated these two ideas to develop a Health Promoting Palliative Care (HPPC) paradigm.

Kellehear also criticized the narrowing of the traditional palliative care approach of medical treatment and the care of dying patients. He thought that palliative care should include the following components: improving self-care and support for patients with chronic or terminal illnesses; providing education and information regarding health, death, and end-of-life care; providing personal and community social support; encouraging rethinking palliative care; and fostering the collaboration between health promotion and palliative care [9]. Currently, palliative care is defined by the WHO as a crucial part of integrated and people-centred health services. It is a global ethical responsibility to relieve severe health-related suffering, be it physical, psychological, social, or spiritual [10].

Since then, hospices and palliative care providers have encouraged medical organizations to accept death as a natural part of the human experience. These ideas had been advocated gradually from the late 1990s to the early 2000s. They have now been expanded to public health and medicine [6]. Professor Kellehear took the lead to bring the concept of health promotion into hospice and palliative care because anxiety, despair, social isolation, social prejudice, social exclusion, family disintegration, unemployment, financial restrictions, spiritual distress or crisis, and even suicide may occur in terminal patients, as well as the surrounding relatives and friends [11]. These are essential aspects that influence quality of life. Many social, psychological, and spiritual issues are too complex for medical institutions to address effectively and become even more difficult in the patients’ final days of life.

A new wave of CC movements is emerging [12,13,14,15]. In 1993, Kozhikode, India, was the first funded CC in Asia. The project included marginalized groups such as the elderly and the mentally ill who were compassionately cared for through community participation. In 2015, the local administration decided to expand its efforts by establishing a platform called Compassionate Kozhikode. Currently, psychiatric institutions receiving frequent community support, community-based rehabilitation plans for chronic mental disorders, and various disadvantaged groups have also been included in Compassionate Kozhikode, with students being permitted to participate [16].

CC in different countries or communities adopts different strategies and develops different programs. The CC in Vic, Spain, embraces positive cultural, social attitudes and activities toward the end of life, provides a comprehensive and integrated care system, and avoids and minimizes suffering [13]. Due to the Medical Assistance in Dying (MAiD) legislation and the growing elderly population, the Canadian government began to provide leadership, education, and promotion programs to help communities across the country understand and establish their CC, with reverence to the local culture to support community members going through the experience [14]. Canada has a national-level promotion organization, e.g., Pallium Canada, a non-profit organization established in 2001 to promote cross-disciplinary palliative care education [15]. Through education courses, the provision of tools and resources, the improvement of the fairness of obtaining palliative care, and the empowerment of front-line workers and non-professional caregivers, the organization promotes the health of the entire population and supports patients and their families by fostering community energy, resilience, and community transformation [17]. The Medical Orders for Life-Sustaining Treatment (MOLST) in New York, a patient-centered shared medical decision-making program, successfully collaborated medical treatment with the community [12]. In Osaka, Japan, a social welfare organization has constructed a community daycare center to help patients manage their daily lives. They expect collaboration from the community and social service departments to help people comprehend the concept of CC, develop problem-solving skills, enhance the experience of dying, the loss of relatives and friends, and meet their welfare needs [11]. According to the above literature review, we have seen CC built by different groups and organizations and different levels of government administration in the context of different cultural and historical backgrounds. However, there is still a lack of discussion of value dissemination, what strategies are adopted, and how people, organizations, and the government should work together.

This project report discusses the CC implementation experiences and preliminary outcomes in the capital of Taiwan, Taipei City. We will describe the CC’s core values and strategies, such as humanistic care, social network, holistic care, and life wisdom. Through specific health care interventions, including home health care services, workshops, conferences, and the Life Issue Cafe in cooperation with a hospital and other social services, a CC has been implemented in the Shilin Old Street neighborhood and later, multiplied in other communities. The following sections explain how we developed the Shilin Old Street community by applying international theory and experience and blending with the local culture, and how we constructed a public–private–partnership framework by connecting government agencies, communities, social welfare institutions, businesses, charitable and religious organizations, and other non-governmental organizations (NGOs).

## 2. Implementation Methods

The CC in Taipei is a significant effort initiated by the city government and a community hospital. It was developed from year-long efforts of community health promotion [18] and health literacy for Shilin Old Street. Professor Allan Kellehear inspired the idea of building Shilin Old Street into a compassionate community when he was invited as a keynote speaker at the Compassionate City Workshop entitled “Age-friendly, dementia friendly, palliative friendly-Supporting a comfortable and independent life for the elderly” in November 2017 hosted by the Health Promotion Administration, Ministry of Health and Welfare (MHW), Taiwan. After a discussion with Professor Allan Kellehear, the administrators of the Taipei City Hospital thought that the Shilin Old Street community was a perfect spot to promote the CC program. Death literacy, which CC promotes [19], can combine with health literacy and become a complete life literacy program that should be provided in community care from life to death.

The MHW began encouraging community health promotion in 1999, facilitating collaboration between the community, public, and private sectors to tackle health problems and improve the quality of life for people in the community. Asset-Based Community Development was then added in 2017 for long-term development, utilizing a resource inventory, emphasizing community participation, and empowering to solve problems and implement strategies to jointly promote the health of community residents.

The Shilin Old Street neighborhood refers to the region surrounding Shilin Shennong Temple, which is currently known as the Jiujia village of Shilin District and was the oldest historically developed place. Located in the centre of the community, Shilin Shennong Temple is a listed historical heritage (since 1812) and folk belief center. Nearby lies the Guo Yuanyi cake museum, a century-old bakery where visitors can learn about history and traditions. Shilin Old Street community represents a mixture of modern and nostalgic culture because of a nearby Mass Rapid Transit (MRT) station. The majority of the population has lived there for decades, with only a handful relocating from elsewhere. According to population statistics (2021), the current population is 6201 (2954 males and 3247 females), with those aged over 65 accounting for 33.29% of the population, which is higher than the Taipei City average of 19.22%, and the national average of 14.05% [20].

A health intervention known as the Shennong LOHAS Station was established through a collaboration between the Taipei City Hospital and the village head. This station is characterized by the local culture and folk religion—where people gather worship regularly—and is integrated with healthcare services. Twice a month, the LOHAS station provides meals, healthcare counseling, and health talks on the first and fifteenth days of the lunar calendar. The nearby Taipei City Hospital Yangming branch also offers home medical care and meals-on-wheels services. The station also works with local NGOs for the disabled and the disadvantaged. All activities included the components of community heritage and culture, e.g., Shennong in folk belief as to the ancestor of Chinese medicine and health seminars with Chinese medicine topic.

By pursuing the components of the International Compassionate City Charter (i.e., public education, community development, health promotion, participatory action, and social ecology), we solidified the core value of the Shilin Old Street CC and formed strategies that suit the reality (Figure 1). We aimed to build a culturally sensitive, sustainable, and holistic CC program by integrating municipal hospital, social, and other services and collaborating with community leaders, NGOs, university students, and volunteers. Innovative programs (such as workshops, conferences, and the Life Issue Café) were presented to enable engagement, public education, and leadership. Furthermore, we organized exhibitions of life aesthetics to raise community awareness of end-of-life issues and explained the evolution of end-of-life care. The four strategies: humanistic care, social network, holistic care, life wisdom, are explained as follows.

### 2.1. Humanistic Care

We hold activities, such as the Festival of Life (death anniversary) to encourage people to share stories of their life, and use events such as Life Aesthetics (exhibition of the artwork or relics of the deceased) to preserve cultural legacy. We also host the Life Issue Café to talk about the issue of life, i.e., health literacy and death literacy. In doing so, we believe mutual trust, empathy, social connection, and solidarity within the community can be rebuilt.

### 2.2. Social Network

We use round table meetings to establish consensus among members of the community, and organize a Neighborhood Watch to improve safety and offer support, such as Meal together, Long Term Care Resources, and Assistant living for the disadvantaged, disabled, and the living alone, thus a friendly environment can be arranged. This can be conducted by collaborating with nearby groups and organizations.

### 2.3. Holistic Care

Our program utilizes home medical care, home pharmacists, home rehabilitation, home care, and home palliative care services of the Taipei City Hospital to offer health promotion and education, screening, prevention, to delay disabilities of the elderly, and increase health and death awareness. As such, people in every stage of life can receive proper support and care.

### 2.4. The Wisdom of Life and Death

The wisdom of pursuing a good life and good death is rooted in the folk. The purpose of palliative care is to promote a good death. We have created the Bucket List Fulfil Project to help the dying and their family to have an opportunity to love, say goodbye, thank, and apologize to each other. We offer Advance Care Planning section to help people finish their Advance Directive, and the home care team in Taipei City Hospital also offers grief counseling and spiritual care. We train community volunteers to care for the disadvantaged population. Pet-friendly services are offered since many people take pets as family members, and the loss of a pet should be considered grief. The Shilin Old Street CC formed a promotion committee and invited professionals, scholars, and social talents to advise the promotion initiatives. Various activities, such as the life issue café, lectures, life story micro-film competition, book clubs, aesthetic activities and exhibitions, drawing competitions, essay competitions, pet adoption reunion and memorial activities, environmentally friendly campaigns, and roundtable meetings, were held to create a trustworthy atmosphere and offer a sense of identity, belonging, and participation.

This community-building program involved many government agencies (Jiujia Village, Shilin District Office, Shilin Police Station, Fire Department Jiantan Branch, Shilin Social Welfare Service Center, Shilin Health Service Center, Shilin Elderly Service Center) and community-related organizations, such as Guo Yuanyi Cake Museum, Guangqing Foundation, Shilin MRT Commercial Association, and Shilinzhuang Culture and History Studio. Students from the University of Taipei, Taipei University of Marine Technology, and Taipei Adventist American School are among the young volunteers recruited by the program.

On 6 April 2018, the Shilin Old Street Compassionate Community was officially launched, and Taipei Mayor Dr. Ko Wenje personally unveiled the plaque. He also declared 6 April as the Taipei International Compassionate Community Day. Professor Allan Kellehear also invited Shilin Old Street CC to join the Public Health Palliative Care International (PHPCI). Currently, 20 villages have joined CC Taipei, and a Taiwan International Compassionate Community Development Association has been formed to promote further initiatives around the country.

## 3. Experience and Preliminary Outcome

Compared with previous CC models around the world, we would like to highlight some significance from our practical experiences, including communication with the end-users, collaboration with stakeholders, and education for service providers.

First, communication with end-users is critical because their understanding of the program and their willingness to be involved is crucial. However, it was considered impolite in traditional Taiwanese society to discuss death directly, especially with the elderly. In order to promote a culture of compassion with one another, especially regarding the issue of death and dying, people need to see that death is a natural part of life, and their death literacy needs to be improved. To establish a supportive network in the community, we first utilized existing social networks and held activities such as seminars, conferences, and the Life Issue Café to facilitate more public engagement.

One of the creative activities we often held was the intergenerational Life Issue Café. We brought the young and the old together to learn through mutual empathy and ex-changes of life experience. These shared learning processes exposed the elderly to social energy and young people, which transformed the notions and beliefs of the older generations and injected the new concepts and ideas from the young while introducing self-identity, life objectives, and interpersonal connections to the elderly to improve community flexibility. The Life Issue Café aided the young to participate innovative thinking and the transmission of experience from the elderly. By participating in community activities, the elderly can reduce cognitive function deterioration through communication and mutual support.

The Life Issue Café is different from the Death Café in previous studies [21] because it does not directly touch the issue of death but instead rationalizes the importance of compassion firstly through discussion and engagement. It is inspired by the Milford Care Centre (MCC)’s Compassionate Communities Project that uses Café Conversation to engage their communities in discussion about death, dying, loss, and care [22]. The method explores collaborative creativity and thought through flexible small group talks. During the debate, one can interact to link diverse viewpoints, listen to reach consensus, think on the problem, and even devise a new course of action. Several life issue cafés have been held (Figure 2), allowing the aged to discuss the theme of life, death, and dying.

University students were invited to join with the elderly to explore these issues. The duration of the meeting is approximately 1.5 h, with each round of dialogue lasting for 20 min. There are three rounds in total, after which collective knowledge is visually generated and the content is shared. The Life Issue Cafe is a simple, adaptable, and efficient way to give people a sense of identification, belonging, and participation in the community in which they reside.

According to our preliminary study, four events were held in 2018, with 27 senior individuals and 60 students attending. The results of two questionnaire surveys showed the attitudes of the senior individuals versus young students toward death: 84.2% vs. 90.9% believe it is vital to address life and death matters with people; 57.9% vs. 45.5% expressed, however, that it is difficult to do so; 78.9% vs. 90.9% believed they have the ability to face death calmly [23].

Second, the collaboration among stakeholders within communities is also important to us. In order to encourage a sense of solidarity and establish a close friendship with all social groups and organizations, we held regular visits to the community care center to teach young students and help them realize the importance of healthy living and end-of-life care. The Shilin Old Street CC worked with university students who visited the elderly in good health, sub-health, disability, and end-of-life and followed-up regularly. Through conversations and activities with the seniors, students engaging in this project learn the fundamental meaning of life, and the elders can adopt healthy views. Community care visits allow college students to access the community, successfully combine school education theory with community practice, and improve social connections and the involvement of the elderly.

Third, we think education for service providers is essential for promoting this new idea of community movement. Succeeding the execution of the tactics mentioned above, we held lots of education and training programs for the hospital staff and community volunteers in order to establish understanding and consensus of the program vision and values. In doing so, community members can actively participate in social lives, share experiences and information, and help one another.

Following in the footsteps of the Shilin Old Street CC, Taipei City has recently developed CC in other administrative districts. The model has been replicated, and there are now 20 compassionate communities spread across the 12 districts of Taipei city. The scope from community participation in end-of-life care to the integration of medical treatment and community resources to improve health care quality. Communities with varying features were created, and each CC had its unique traits, e.g., the Huxing village CC in Neihu District with elderly beliefs, the Hulu village CC with a pet-friendly environment, the Changchun village CC with friendship, and other districts with their own regional characteristics.

## 4. Discussion

### 4.1. Glocalization of CC

Glocalization refers to a fusion of globalization and localization, signifying an increase in the relevance of continental and global levels, as well as an increase in the importance of local and regional levels, reflecting both local and global considerations [24]. According to Robertson, the genuine process of globalization includes the localization process within certain conditions and contexts. Robertson advocated a dual and mutual process that transforms localized models into global trends and goals [25]. When global universality is applied to other locations, it assimilates local traits and conditions and becomes locally exclusive. We believe the CC development is a glocalization process: theory, practice guidelines, and standards are distributed over an international network, while each CC was formed to meet local culture and community conditions.

### 4.2. Collaboration, Partnership, and Integration

The majority of CC organizations develop alliances by combining various types of public and private organizations, including those of education, medical care, social services, culture, religion, volunteer, business, and non-profit organizations. All fields of health and social care professions, such as palliative care, homes, communities and medical care, long-term care, physicians, and social workers, need to participate. All partners must work together through open communication and a shared vision, whichever launches the CC.

In Taipei, the CC evolved from a health promotion background and merged with palliative care. We believe in holistic care that integrates the personal needs for physical, psychological, and social well-being and the care needs ranging from healthy people to palliative care patients. The convergence of well-being and palliative care is critical for the entire community.

### 4.3. Communication and Public Education

A good communication plan is required to make people aware of the importance of open discourse concerning palliative care and death literacy. It is critical to boost communication and educate the people on the relevance of the government policies for end-of-life care. Communication is essential for a successful collaboration to achieve the same aims and speed up the promotion of CC. As people have taboos about dying, death, and grieving, and they are hesitant and fearful of addressing death, public education is critical to changing society, forming a culture of compassion, and sharing care obligations towards the end of life. There are also difficulties, such as a reluctance to ask for or accept help from others and provide assistance. Education should extend beyond campus and communities to improve social connectivity and death literacy. Professionals such as physicians, nurses, and caregivers may also need to incorporate death education into their curricula as dying, death, and mourning are not currently taught [4,26]. People of all ages and backgrounds must confront loss and experience death and bereavement, necessitating the provision of death and grief education.

### 4.4. Local Culture and Religions

Care, dying, death, and grieving occur in all aspects of our daily lives and human interactions, and they have always been impacted by local culture and religious beliefs [27]. Cultural or religious beliefs may influence one’s reactions and subsequent behaviors [28]. It may help one to understand the difficulty of caring for the loved one who is seriously ill, recognize the cycle of life, and affirm the spirit of CC. Partnerships with religious groups are viable to create a compassionate and caring community [29]. Similar to other temples and Christian churches, the Shennong Temple helps promote palliative care and end-of-life education in the community, encourages its followers to cultivate and exercise compassion, and grasps the vocabulary and practices of palliative care.

### 4.5. COVID-19 Pandemics and CC

The outbreak of the COVID-19 pandemic has brought increased death and dying into communities, requiring new demands that significantly alter the social engagement patterns. Everywhere, social care and volunteering halted because they are deemed too dangerous. Despite this, people still want to connect, take action, and support those in need. The CC in Taipei is well-established and has not stepped back during the COVID-19 pandemic but continues to support its community members and presents strong solidarity and support to the health professionals. According to the advocate posted on the Public Health Palliative Care International website, compassionate streets and neighborhoods can respond to the COVID-19 pandemic by mobilizing community helpers, providing practical assistance and emotional support, and building local communications through information technology. Providing both physical and mental support is a demand of the heart and our compassionate response in the time of adversity brought by the COVID-19 pandemic [30]. CC can offer established support and social connections to those dealing with death, dying, loss, and caring.

## 5. Conclusions

CC has actively created a sense of solidarity, tranquility, and enjoyment in the community by capitalizing on the need for aged care and ageing in place. This strategy acquires worldwide and local support and collaboration with the medical teams, universities, and NGOs. We believe that the government and people working together can foster a friendly, caring community for the elderly and form a caring community model from birth to death.

This project report presents a paradigm of collaboration that combines health promotion, palliative care, and community assistance. Many education and training courses and community-building activities were conducted using a culturally sensitive approach, and appropriate health and end-of-life care were provided. The collaboration between hospital and community groups has resulted in positive and meaningful experiences for caregivers and patients. Community involvement lessens the workload of the hospital palliative care team and offers further service quality improvement.

The tentacles of palliative care and service can be extended to more persons in the community who have terminal diseases or require palliative care through activities. An ideal CC is a collaborative effort of caregivers, family members, friends, neighbors, volunteers, and the palliative care staff. The general public has diverse perspectives on death, dying, and grieving due to the diversity of cultural background, religious beliefs, and life experience. These are commonplace experiences for most people but may be difficult for medical experts to comprehend. The demands and difficulties necessitate community participation to prioritize the well-being of all community members. As a result, strengthening the collaboration is a viable option for CC.

Our practical experience has shown that empowering the community may effectively improve their capacity to deal with death, dying, and bereavement issues. By forming alliances and empowering the community, we can help bereaved families receive care from their community and eventually return to regular life. Spiritual care in our CC has become a regular topic of conversation among community members. The Shilin Old Street CC’s experience intends to motivate more government agencies, hospitals, and communities to participate and improve the overall quality of end-of-life care. This culturally sensitive, sustainable, and holistic CC model bridges the bereavement support gap in palliative care by combining the municipal hospital, social, and other services and partnering with community leaders, NGOs, university students, and volunteers. Our intergenerational learning activity between the young and old is novel; it improves the sense of belonging of the elderly and creates a generation-friendly community. Participation of young students also allows them to apply what they have learned and return the favor to society, thus developing intergenerational reciprocity.

## Figures and Tables

**Figure 1 healthcare-10-00177-f001:**
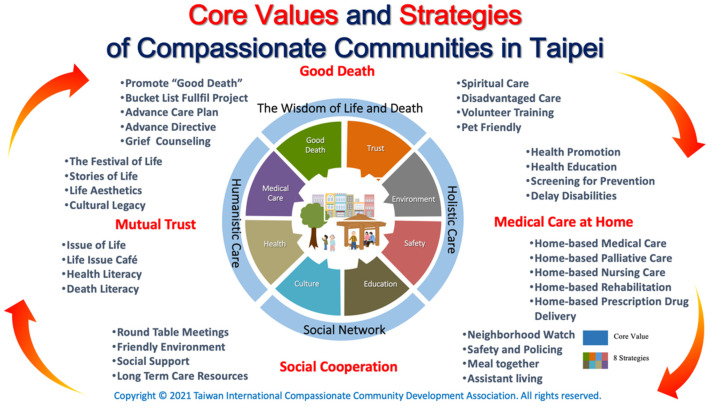
Core values and strategies of compassionate communities in Taipei.

**Figure 2 healthcare-10-00177-f002:**
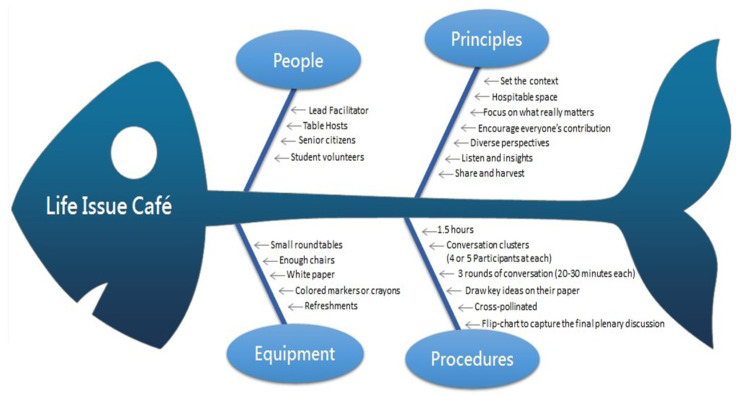
Method of Life Issue Café.

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
