# Peer review of "Implementation of Compassionate Communities: The Taipei Experience"

_healthcare, 2022, doi:10.3390/healthcare10010177_

Round 1
Reviewer 1 Report
The article “Implementation of compassionate communities: The Taipei Experience” describes the core values and strategies of compassionate communities in Taipei: humanistic care, social network, holistic care, and life wisdom. Through certain health care interventions including home health care services, workshops, conferences, and the “Life Issue Café” in cooperation of a hospital, social, and other services a CC is implemented in the “Shilin Old Street neighbourhood” and later in other communities. The article covers the topic of the implementation of CCs, which becomes increasingly important in meeting the needs of the dying in the developed world, as the demographic change will lead to more elderly in the future.
The major concern that the article lacks central components of a research study has been addressed in restructuring the paper as a “Project Report” (https://www.mdpi.com/about/article_types); most of the minor concerns have been addressed as well.
Only minor concerns preclude publication at this stage:
- Please rename from “Article” (p.1, l.1) to “Project Report”.
- Introduction:
- Palliative care is defined by the WHO … ("care" is missing, p.2, l.55)
- Implementation Methods
- Number missing in date (p.3, l.121)
- (…) education, screening, prevention, and (…)? p.4, l.181
- Figure 1, figure legends: Core values and strategies (…)
- Bucket List Fullfil Project
- A better resolution should be used to reduce pixeling
- Discussion
- We believe in holistic care that integrates the personal needs for physical, psychological, and social well-being and the care needs ranging from healthy people? to palliative care patients? (p.7, l. 313-315)
- According to the advocate posted on the PHPCI (what does this acronym mean?) website, compassionate streets and neighborhoods can respond to the Covid-19 pandemic? by mobilizing (…)
- Conclusion:
- “This article” (p.8, l.366) should be changes to “Project Report”
- References: The references have to be checked carefully, e.g. ref. 23 (Roudometof, V.,"Theorizing glocalisation") is used in conjugation with the data of a preliminary study conducted by the authors -ref. 22?
Author Response
Dear Reviewer:
Many thanks for the comments. We have revised our manuscript according to your comments and using the “Track Changes” function to marked up all revisions made to the manuscript. Please also find the following table of our response to each comment.
All the best and many thanks,

Reviewer 2 Report
Nice work with the edits!
Author Response
Dear Reviewer:
Many thanks for the comments.
All the best and many thanks,

Reviewer 3 Report
The manuscript has been substantially improved a lot. Just few minor comments:
1. The article type in the manuscript should be changed into "Concept paper" rather than an "Article."
2. Please also ensure if the figure is the one developed by the authors and no copyright issues.
3. You cited 31 references in the text, but only 30 references are in the list. Also, please conform to MDPI reference style.
Author Response
Dear Reviewer:
Many thanks for the comments. We have revised our manuscript according to your comments and using the “Track Changes” function to marked up all revisions made to the manuscript. Please also find the following table of our response to each comment.
All the best and many thanks,

This manuscript is a resubmission of an earlier submission. The following is a list of the peer review reports and author responses from that submission.
Round 1
Reviewer 1 Report
See file.

Reviewer 2 Report
It is a significant effort to initiate Compassionate Communities in the government and hospital level. It is worthwhile to share the experience to other health care provider or government especially in current COVID-19 pandemic time. I suggest accepting the article after modifying the English grammar.
Reviewer 3 Report
HEALTHCARE: Integration, partnership, and cultural sensitivity: Implementation and preliminary outcomes of public health palliative care initiatives of compassionate communities in Taipei City
This study sought to evaluate a community program to support community members at end of life, specifically the “Compassionate Communities”. Overall, the topic is important in understanding how to best support community members at end of life. However, there are several limitations that must be addressed. In general, a thorough review of the text needs to be conducted to ensure proper grammar and sentence structure. Additionally, the authors need to ensure that all abbreviations have been clearly defined throughout. In its current form, this paper is very hard to follow and missing important information to fully understand the purpose and methods, and therefore, does not meet the standards for publication.
Abstract
- Stronger introduction to establish why this topic is important
- Please add information about how feasibility, social connectedness, and participant rates were measured (i.e., time period, measurement tool, etc…).
- Add information about the sample of individuals this was evaluating
Intro
- Please address incomplete sentences. For example, “A supportive environment for palliative care and grief counseling” feels out of place and incomplete.
- The whole first paragraph feels very disjointed. I had to read multiple times to follow what the authors were trying to explain.
- The whole paragraph discussing prevention, harm reduction, and early intervention seems unnecessary since this paper is targeting end of life/palliative care.
- Set-up the paragraph where you discuss the programs around the world differently. For example, “Currently, multiple community movements aimed at improving end of life care exists around the world.” Then dive into each program. Also, keep information regarding each country together. You talk about Canada, then to New York, then back to Canada.
- I recommend starting with Kozhikode, India since it is the first compassionate community then discuss the others in order of development.
- Need to discuss the gaps in literature/current programs and why another model is needed (i.e., yours). Why don’t you just use one of the other established program models?
Methods
- The first paragraph reads as more of an introduction paragraph, not a materials and methods. You should explain the necessity for the program in the introduction.
- Figure needs to be reconstructed. It is hard to follow and read. Definitely needs a footnote to explain what is going on.
- Remove the history of the area, not necessary and adds confusion.
- You say you built and replicated- what did you replicate?
Due to the lack of clear writing and substantial short-fallings in information regarding methods, the rest of the manuscript was not reviewed.
Reviewer 4 Report
- This paper should be restructured to a concept paper.
- The subsection that follows the conventional style of original research reporting is not appropriate.
- I would like to see further debates on the significance of the study comparable to some theoretical models or conceptual frameworks to the subject matter.
- Discuss the gaps from previous theoretical frameworks and the significance of the current one.
- The paper was conceptualized based on a consensus-driven approach from Kellehear. Was there a notion that recommended integration of palliative care during pandemic times? This needs to be discussed.
- I would like to see some practical public health implications that could be derived from this concept paper - to stakeholders, to the end-users and service providers.
- Public health mitigation and suppression strategies was the focus to tackle the COVID-19 pandemic. Could this be a confounder or mediating effect or a barrier to provider of the palliative care service to the patients? How was this issue tackled?